# Are (Co)Polymers of 1,1,3,3,3-Pentafluoropropene Possible?

**DOI:** 10.3390/molecules28124618

**Published:** 2023-06-07

**Authors:** Frédéric Boschet, Georgi Kostov, Hristina Raynova, Bruno Améduri

**Affiliations:** Institut Charles Gerhardt, CNRS, ENSCM, University of Montpellier, 34095 Montpellier, France; fboschet@hotmail.com (F.B.); georgikostov43@gmail.com (G.K.); hristina.raynova@enscm.fr (H.R.)

**Keywords:** copolymerization, 1,1,3,3,3-pentafluoropropene, vinylidene fluoride, chlorotrifluoroethylene, 3,3,3-trifluoropropene, hexafluoropropene, 3-isopropenyl-α,α’-dimethylbenzyl isocyanate, vinylene carbonate, perfluoromethylvinyl ether, *tert*-butyl 2-trifluoromethacrylate

## Abstract

The copolymerization and terpolymerization of 1,1,3,3,3-pentafluoropropene (PFP) with various combinations of fluorinated and hydrogenated comonomers were investigated. The chosen fluoromonomers were vinylidene fluoride (VDF), 3,3,3-trifluoropropene (TFP), hexafluoropropene (HFP), perfluoromethylvinyl ether (PMVE), chlorotrifluoroethylene (CTFE) and *tert*-butyl-2-trifluoromethacrylate (MAF-TBE), while the hydrocarbon comonomers were vinylene carbonate (VCA), ethyl vinyl ether (EVE) and 3-isopropenyl-α,α-dimethylbenzyl isocyanate (m-TMI). Copolymers of PFP with non-homopolymerizable monomers (HFP, PMVE and MAF-TBE) led to quite low yields, while the introduction of VDF enabled the synthesis of poly(PFP-*ter*-VDF-*ter*-M_3_) terpolymers with improved yields. PFP does not homopolymerize and delays the copolymerizations. All polymers were either amorphous fluoroelastomers or fluorothermoplastics with glass transition temperatures ranging from −56 °C to +59 °C, and they exhibited good thermal stability in air.

## 1. Introduction

Pentafluoropropene (PFP) and 3,3,3-trifluoropropene (TFP), both produced by Great Lakes/Chemtura, have barely been studied either in telomerization [1] or polymerization. TFP was first synthesized by Haszeldine [2]. It is used as precursor of the fluorinated silicone marketed by Dow Corning under the Silastic^®^ tradename. This poly(trifluoropropyl methyl siloxane) has very interesting properties, such as inertness to chemicals, low surface tension and low glass transition temperature (ca. −70 °C) [3]. On the other hand, for PFP, many fewer studies have been reported. PFP exists in two isomers: 1,1,3,3,3-pentafluoropropene (2H-PFP) and 1,2,3,3,3-pentafluoropropene (1H-PFP). The radical cotelomerization of 2H-PFP in the presence of various co- and ter-monomers was described [4]. Regarding (co)polymerization, only a few reports can be summarized in the following lines. Originally, in the 1950–1960s, Dupont (Wilmington, DE, USA) [5,6] and Kellogg Co. (Battle Creek, MI, USA) [7] investigated the preparation of fluorinated elastomers, and they obtained thermostable, oil-resistant elastomers that were used as compactors in aggressive media. It is well understood that elastomers based on vinylidene fluoride (VDF) with either hexafluoropropylene (HFP) or tetrafluoroethylene (TFE) are susceptible to bases. Thus, to circumvent this issue, the Montecatini Edison Company (Milan, Italy) [8,9] developed new fluorinated elastomers based on VDF and 2H-PFP which are thermostable and solvent-resistant compared to the co- and terpolymers made of HFP. The first fluorinated copolymers based on PFP were patented by Sianesi and Caporiccio [10]. These copolymers were produced using titanium tetraisopropylate/dichloromethane/aluminum triisobutyl (Al/Ti ratio = 2) as the initiator system at 40 °C for 200 h in 8%, yield showing its quite low PFP reactivity. The patent also claimed the copolymerization of PFP with VDF, vinyl fluoride, trichloroethylene and other fluorinated olefins. Sianesi et al. [8] reported that poly(VDF-*co*-2H-PFP) copolymers could be easily molded and vulcanized. They were produced using organic peroxides as initiators (in 0.01–2.00 wt%) either in aqueous suspension or in solution. When the PFP amount was less than 5–15 mol%, thermoplastics were obtained, whereas in the 10–70% range, elastomers were prepared. Crosslinked fluorinated elastomers could be obtained using aliphatic polyols and organic peroxides. The same authors [8,9] also reported a terpolymer based on VDF and 1H- and 2H-PFP in similar conditions. In addition, Bolstad [11] patented the copolymerization of fluorinated olefins that can be divided into three groups according with their volatility. The second group contains fluoroolefins with a maximal pressure of 5 bars at room temperature, which include PFP, but the patent lacks data regarding its copolymerization. In 2001, Dupont Dow Elastomers LLC [12] disclosed the radical copolymerization of 2H-PFP with various fluoroolefins, using PFP as a cure site monomer. The copolymerizations were initiated using organic peroxides in aqueous media. The molar mass of the resulting elastomers ranged between 50,000 and 2,000,000 g·mol^−1^, which could be regulated using a suitable chain transfer agent. It was shown that the PFP units (0.3–3 mol%) were randomly distributed along the chains. Unfortunately, and as expected, this rather old patent literature does not bring much information on the accurate experimental synthetic conditions and characterizations of such copolymers. The following year, another patent was granted, claiming the terpolymerization of PFP (as a cure site monomer) with ethylene, tetrafluoroethylene and perfluoroalkyl vinyl ethers by a continuous emulsion process to produce a base-resistant fluoroelastomer [13]. From the example given, the 2H-PFP feed content of 5.8 wt% yielded a terpolymer containing only 1.0% of 2H-PFP and an overall conversion of 77.6%. When 2H-PFP was replaced by trifluoropropene (TFP), the incorporation of TFP was better than that of 2H-PFP, with similar amounts of the other monomers and conversion, as PFP went from 1.2 wt% in the feed to 1.8 wt% in the terpolymer. In 2015, Daikin Industries Ltd. claimed the use of 1H or 2H PFP to improve the processability and moldability of polytetrafluoroethylene (PTFE) thanks to a core–shell structure [14]. The modified PTFE particles have a core–shell structure that includes a particle core and a particle shell; the particle core has repeating units of tetrafluoroethylene and a perfluoroalkyl ethylene (C_1_-C_10_), and the particle shell has repeating units of TFE, a (perfluoroalkyl)ethylene and 1H or 2H PFP. To achieve these core-shell structures, PFP was added in small amount after 75% or 90% conversion of TFE and yielded 0.010 to 0.024% of PFP in the resulting particles (although claimed to be 0.05%). No indication was given of the conversion or molar masses.

The purpose of this article aims to investigate the radical copolymerization of 1,1,3,3,3-pentafluoropropene (2H-PFP) in the presence of various commercially available fluorinated and non-fluorinated comonomers to study them thoroughly (hence highlighting their microstructures) and to supply their thermal properties. Furthermore, co- and terpolymerization of PFP were attempted to circumvent strategic routes involving HFP, to find an alternative to this fluoromonomer.

## 2. Results and Discussion

This section is divided into two parts related to copolymerizations and terpolymerizations involving PFP.

### 2.1. Radical Copolymerization of PFP

The radical copolymerizations of PFP with vinylidene fluoride (VDF) and with *tert*-butyl 2-trifluoromethacrylate (MAF-TBE), and the terpolymerization of PFP with MAF-TBE and third comonomer M_3_, were achieved according to the conditions indicated in Table 1.

#### 2.1.1. Radical Copolymerization of PFP with VDF

Usmanov et al. [15] synthesized fluoroelastomers from the radical copolymerization of 2H-PFP with VDF initiated by γ-rays (Figure 1).

Sianesi and Caporiccio [10] also reported the copolymerization of VDF with PFP initiated by organic peroxides and metallic complexes catalysts. The copolymer was obtained in low yield (<10%). Furthermore, Usmanov [15] assessed the reactivity ratio for the copolymerization of VDF with PFP initiated by electron beam (*r*_PFP_ = 0.06 and *r*_VDF_ = 9.0 at 30 °C), showing a quite poor homopropagation of PFP, as expected. Later, Otazaghine et al. [16] studied the kinetics of radical copolymerization of VDF with CF_2_=CH-C_6_F_13_ initiated by *tert*-butyl peroxypivalate and determined their reactivity ratios: *r*_C8F15H_ ≈ 0 and *r*_VDF_ = 12.0 at 74 °C.

We carried out the reactions in conditions as described above in the experimental section. Only a small amount of copolymer was obtained in this case in low yield (11%, Table 2). This result is in agreement with the literature [15,16]. The microstructure of the resulting poly(PFP-*co*-VDF) copolymer indicates the formation of PVDF micro-blocks that display both a regular structure ~CH_2_CF_2_-CH_2_CF_2_~ (head-to-tail additions) and reverse additions ~CH_2_CF_2_-CF_2_CH_2_-CH_2_CF_2_~ (head-to-head and tail-to-tail additions) [17,18,19]. This was evidenced by a signal located at −91 ppm in the ^19^F NMR spectrum of the copolymer (Figure 1) assigned to the normal additions and two peaks centered at −113 and −116 ppm attributed to the reverse head to head one [20,21,22].

The presence of PFP units is evidenced by the signals located at −60.0, −65.3 and −66.9 ppm assigned to CF_3_ group of PFP. The molar percentages of VDF and PFP were calculated using the following equations:$$\%VDF=\frac{\left(I_{-91}+I_{-93}+I_{-110}+I_{-116}\right)/2}{\left(I_{-91}+I_{-93}+I_{-110}+I_{-116}\right)/2+\left(I_{-60}+I_{-65}+I_{-67}\right)/3}\times 100$$
$$\%PFP=\frac{\left(I_{-60}+I_{-65}+I_{-67}\right)/3}{\left(I_{-91}+I_{-93}+I_{-110}+I_{-116}\right)/2+\left(I_{-60}+I_{-65}+I_{-67}\right)/3}\times 100$$

Copolymerization results are listed in Table 1. As expected, the VDF contents in the copolymers were higher than those in the feed ratio (62 vs. 20 mol%, Run No 1, Table 1), confirming that VDF is much more reactive than PFP (for which content in the copolymer reduced from 80 mol% in the feed to 38 mol%). The yield of the reaction was also low due to the low PFP reactivity compared to that of VDF, and we observed a random distribution of a PFP unit between VDF micro-blocks.

The ^1^H NMR spectrum of the poly(PFP-*co*-VDF) is presented in Appendix A in the electronic supporting information (ESI) and exhibits the intense signal characteristic of normal VDF additions. This signal is located in the 2.8–3.0 ppm range [15,16,17], while that at 3.1 ppm is attributed to the methyne proton in PFP. Peaks at 1.4 and 1.8 ppm are assigned to initiator moieties in the polymer chain (normal and reverse addition, respectively) arising from the direct initiation from the methyl radical [20].

#### 2.1.2. Radical Copolymerization of PFP with MAF-TBE

To the best of our knowledge, that reaction has never been reported, probably because these monomers cannot homopolymerize separately and are both electron-withdrawing. The radical copolymerization of PFP with *tert*-butyl 2-trifluoromethylacrylate (MAF-TBE) was carried out using two different initial molar ratios: 50/50 or 80/20. The results are listed in Table 1. When acetonitrile was used as the solvent, no significant yield difference was observed (14% vs. 10%). However, when carried out in 1,1,1,3,3-pentafluorobutane, the yield increased two-fold (26% vs. 10%) when increasing the PFP amount. Thus, fluorinated butane was the preferred solvent for the terpolymerization (see Table 1). After purification, the different copolymers were characterized by NMR spectroscopy.

The ^19^F NMR spectra (Appendix A in ESI, Entry #5) exhibits the characteristic signals observed at −60 and −61.5 pm assigned to the –CF_3_ group of PFP while the two intense peaks centered at −68 ppm are attributed to the –CF_3_ of MAF-TBE unit. The signals at −72.5 and −75.7 ppm were assigned to the non-equivalent fluorine atoms of CF_2_ from PFP. The PFP molar content was calculated from I_−60/_(I_−60_+ I_−68_), where I_j_ corresponds to the integral of the signal at j ppm. The composition of the copolymer is composed of 99 mol% of MAF-TBE.

The ^1^H NMR spectrum (Appendix A in ESI) shows a singlet centered at 1.5 ppm assigned to the *tert*-butyl group of MAF-TBE units. Signal centered at 2.5 ppm was assigned to the methylene groups in MAF-TBE [23] units, whereas that centered at 5.1 ppm was attributed to the methyne protons of PFP (borne by an asymmetric carbon atom) [24].

The analyses of the obtained copolymer indicate a molar mass of ca. 2000 g·mol^−1^ (see Table 3), which should correspond to nine MAF-TBE units and one PFP unit, which is unexpected as MAFTBE is known not to homopolymerize [1].

#### 2.1.3. Radical Copolymerization of PFP with HFP

As above, HFP does not homopolymerize [25,26,27], and its copolymerization with PFP has never been reported. Thus, the radical copolymerization of PFP with HFP was carried out in C_4_F_5_H_5_ initiated by Trigonox^®^ 101 at 130 °C for 20 h. The resulting poly(PFP-*co*-HFP) was obtained in very low yield (3%, Table 2), characterized as follows.

The ^19^F NMR spectrum (Appendix A in ESI) exhibits the signals centered at −60 and −65 ppm attributed to the –CF_3_ group in PFP, and characteristic signals of HFP (trifluoromethyl group −70 and −75 ppm) [4,20,21,22]. Other signals assigned to HFP are noted at −120 ppm and −130 ppm (-CF_2_-), and −175 ppm and −185 ppm (-CF-). The signal of the CF_2_ of TFP is located at −100 ppm. The signal located at −210 ppm, as the signature of a CFH group [28] arises from the transfer to solvent.

As above, the product is a viscous liquid and, accordingly, the molar mass assessed by NMR or SEC is quite low (1300 and 1600 g·mol^−1^, respectively), thus yielding a cooligomer. It can then be deduced that the incorporation of PFP significantly lowers the yield, and the incorporated amount is only 17 mol% (compared to 69 mol% in the feed). It is thus deduced that PFP is about 6 times less reactive than HFP [27]. As for poly(PFP-*co*-MAFTBE) copolymers, it is concluded that the obtained oligomers should consist of about four units of HFP with one unit of PFP, suggesting that, under these conditions, HFP tends to oligomerize [24], which is quite surprising [27].

#### 2.1.4. Radical Copolymerization of PFP with PMVE

Then, the radical copolymerization of PFP with perfluoromethyl vinyl ether (PMVE), another non-homopolymerizable fluoromonomer [29,30], has been attempted to be copolymerized with PFP. The results are listed in Table 2.

It is worth noting that the copolymerization of PFP with PMVE yielded the highest PFP content in the copolymers (compared to all above copolymerizations): 33 mol%.

The ^19^F NMR spectrum (Appendix A in ESI) exhibits the signals centered at −50 and −59 ppm attributed to the –CF_3_ group in PMVE and PFP, respectively. The peak at −146 ppm was assigned to the -CF- fluorine of PMVE [25,26]. The two fluorine of CF_2_ in PMVE were evidenced by the signals centered at −116 and −124 ppm and the multiplet in the −133 to −142 ppm range. The CF_2_ groups in PFP also form a complex structure in the −96 to −110 ppm range.

The ^1^H NMR spectrum (Appendix A in ESI) exhibits the signal of the methyne protons of PFP at 4.8 ppm as well as peaks resulting from direct initiation at 1.0 and 1.4 ppm.

With a glass transition temperature (T_g_) of −56 °C, the poly(PFP-*co*-PMVE) cooligomer is among the lowest from all the elastomers synthesized above, and the low molar mass might explain such a low value.

#### 2.1.5. Radical Copolymerization of PFP and TFP

Starting from 75 mol% of PFP (Run 9, Table 2), the radical copolymerization of PFP with TFP yielded a cooligomer bearing a high PFP amount (19 mol%), as for PMVE. The ^19^F NMR spectrum (Appendix A in ESI) displays the characteristic signals of TFP (trifluoromethyl peaks centered at −60 and −64 ppm) [31,32,33,34] and that of PFP at −66 and −70 ppm, while the ^1^H NMR spectrum (Appendix A in ESI) exhibits the three multiplets of the protons borne by asymmetric carbon atoms of TFP and PFP located at 4.7, 4.0 and 3.4 ppm, respectively. The complex signal centered at 2.2 ppm is assigned to the methylene protons of TFP as evidence of the AB system in the ABX system.

As for the cooligomers based on PFP and PMVE, this poly(PFP-*co*-TFP) cooligomer exhibits a very low glass transition temperature (−53 °C).

Regarding additional thermal properties, Figure 2 plots the thermal stability of the copolymers and terpolymers based on PFP, VDF and a third comonomer versus the molar mass assessed by size exclusion chromatography or NMR. As expected, these data show that the thermal stability of the copolymers increases with molar mass.

It was also worth plotting the evolution of the thermal stability versus the comonomer content in the copolymer. Figure 3 indicates that the thermal stability increases with increasing VDF content but decreases with either HFP or PFP units in the copolymers. As shown above, this feature might be linked to the molar mass, since VDF yielded higher ones than HFP or PFP did.

#### 2.1.6. Radical Copolymerization of PFP with CTFE

Chlorotrifluoroethylene (CTFE) is well known to yield alternating copolymers with electron-donating monomers such as vinylene carbonate (VCA), vinyl ethers or m-TMI [1,35,36,37].

The radical copolymerization of PFP with CTFE also shows that PFP is less reactive than other fluorinated monomers, as the PFP amount in the copolymer is much lower than that in the feed (1.5% vs. 68.8% Run 1 Table 4). Furthermore, the copolymer was also obtained in poor yield (6%), indicating that the polymerization is not strongly favored. Once again, the concept of termonomer-induced copolymerization (TIC) (process pioneered by Weise [38] in 1971) can be applied to this copolymerization to enhance the incorporation of PFP in the copolymer and increase the yield (see Section 5: Radical terpolymerization of PFP with CTFE and M_3_ monomer).

After purification, the resulting poly(PFP-*co*-CTFE) copolymer was characterized by ^19^F NMR spectroscopy. The spectrum (Appendix A in ESI) displays the expected signals located at −60 ppm as evidence of PFP units and the characteristic signals assigned to CF_2_ (non-equivalent fluorine atoms inducing a broad signal centered at −107 ppm and the CFCl at –126 ppm. An additional small one at −100 ppm is assigned to the CF_2_ group in CTFE adjacent to PFP unit (i.e., CTFE-PFP dyad). Finally, the signal at −127 ppm is assigned to the CFCl in the CTFE-CTFE dyad [39], while that at −151 ppm is attributed CFCl in -CF_2_CFClH end group that results from the transfer to the solvent [40]. The ^1^H NMR spectrum (Appendix A in ESI) shows the expected signal at 3.6 ppm for the methyne proton of PFP and a peak at 1.4 ppm that was assigned to chain ends from the initiator (methyl protons).

Independently from the small PFP amount, the poly(PFP-*co*-CTFE) is soluble in THF, allowing the determination of its molar masses by size exclusion chromatography (8500 g·mol^−1^).

#### 2.1.7. Radical Copolymerization of VDF with MAF-TBE

In this case of copolymerization, the yields were much higher (twice as much for Entry #3, Table 1) than for the (PFP;VDF) or (PFP;MAF-TBE) couples because VDF and MAF-TBE is a very reactive monomer pair [41], as evidenced by previous reports on the copolymerization of VDF with 2-trifluoromethacrylic acid H_2_C=C(CF_3_)CO_2_H (MAF) [25,42]. From this copolymerization, two fractions were obtained, probably linked to their crystallinity: the first one (16 wt%) contains a high VDF content (92 mol%), while the second fraction (84 wt%) consists of an almost alternating poly(VDF-*alt*-MAF-TBE) copolymer in 43/57 mol%. The ^19^F NMR (Appendix A in ESI) shows signals located between −68 and −71.5 ppm, assigned to the trifluoromethyl groups of MAF-TBE [41]. Characteristic signals of VDF units are located at −91 ppm (-CH_2_CF_2_CH_2_CF_2_-, normal head-to-tail addition) and −94 ppm (–CH_2_CF_2_CH_2_C(CF_3_)(CO_2_-tBu), with an alternating dyad [42,43,44,45,46]. We also evidenced the presence of some reverse VDF addition (signals at −113 and −116 ppm as explained above). The VDF molar content was calculated according to the following equation:$$I_{VDF}=\frac{(I_{-91}+I_{-94}+I_{-113}+I_{-116})/2}{(I_{-91}+I_{-94}+I_{-113}+I_{-116})/2+(I_{-68}+I_{-71})/3}$$
where *I_j_* represents the integral of the signal centered at *j* ppm.

The observed alternating structure is in agreement with the work of Souzy et al. [25], which indicated that the radical copolymerization of VDF with MAF with a VDF feed content ranging from 25 to 75% led to an alternating poly(VDF-*alt*-MAF) copolymer. Above 75% VDF in the feed, PVDF micro-blocks can also be found in addition to the alternating structure. The PVDF micro-blocks are responsible for the peaks centered at −91 ppm but also those related to the reverse head-to-head VDF additions (−113 and −116 ppm), found in very small intensities in the present case.

The ^1^H NMR spectrum of poly(VDF-*co*-MAF-TBE) copolymers (Appendix A in ESI) evidences the presence of head-to-tail VDF-VDF additions by the signal ranging between 2.7 and 3.2 ppm, while those assigned to reverse tail-to-tail VDF-VDF additions are located at ca. 2.4 ppm. Signals corresponding to MAF-TBE are located at 1.2 ppm (methyl in *tert*-butyl group) and 2.4 (for the methylene protons overlapping with reverse addition of VDF). The signal at 0.8 ppm was assigned to direct the initiator moiety.

#### 2.1.8. Radical Copolymerization of VDF with HFP

Hexafluoropropylene (HFP) is often used as a comonomer in the copolymerization of fluorinated olefins (especially VDF) [47] to increase the chain flexibility. It leads to commercially available poly(VDF-*co*-HFP) copolymers such as Solef^®^, Tecnoflon^®^, Kynar^®^, Daiel^®^, Viton^®^ and Dyneon^™^, etc. [26,47,48], tradenames, marketed by Solvay Specialty Polymers, Arkema, Daikin, Dupont (now Chemours) and 3M/Dyneon, respectively.

When copolymerizing VDF with HFP, two behaviors of polymers can be obtained: thermoplastic and elastomeric poly(VDF-*co*-HFP). If the HFP content in the poly(VDF-*co*-HFP) copolymer is above 15 mol%, soft amorphous products are obtained [1,49,50,51,52].

In comparison, we investigated the copolymerization of VDF with HFP in similar conditions as described above for PFP with VDF or HFP. Results are shown in Table 2. The (VDF;HFP) couple led to a copolymer obtained in high yield (87 wt%).

NMR characterization and attribution of the poly(VDF-*co*-HFP) copolymers are available in the supporting information (Appendix A) [53]. The ^19^F NMR spectra (Appendix A in ESI) shows intense signals centered at −91, −94 and between −108 and −116 pp, assigned to both normal and reverse additions of VDF. The peak located at −70.5 ppm and the doublet centered at −75.4 ppm were assigned to the trifluoromethyl of HFP. The signal centered at −118.5 ppm was assigned to the -CF_2_- of HFP, as well as these at −180.5 and −184.2 ppm to the >CF- of HFP. Furthermore, the signal at −103.4 ppm in the HFP-VDF-HFP triad structure, –CF_2_CF(CF_3_)-CH_2_CF_2_-CF_2_CF(CF_3_), confirms the observations of Moggi et al. [54]. The ^1^H NMR spectrum (Appendix A in ESI) exhibits the presence of multiplets in the 2.1–2.4 ppm and 2.6–3.4 ppm ranges assigned to the methylenes of VDF from the reverse and normal additions, respectively.

### 2.2. Radical Terpolymerization of PFP with VDF and Fluorinated M_3_ Monomer

#### 2.2.1. Radical Terpolymerization of PFP with VDF and MAF-TBE

The terpolymerization (Figure 2, Table 1) was carried out with various monomer ratios, and were initiated by Trigonox^®^ 101 and DTBP in 1,1,1,3,3-pentafluoropropane. As indicated above, the literature [25,41,55,56] reports that the copolymerization of VDF with MAF-TBE leads to alternating poly(VDF-*alt*-MAF-TBE) copolymers in good yields. It is expected that the incorporation of PFP in the terpolymerization could be favored according to the termonomer-induced copolymerization [38,57].

The terpolymerization of PFP with VDF and MAF-TBE (Table 1, Entries #6–10) indicates the presence of an alternating structure between VDF and MAF-TBE, in agreement with the literature [25,41,55]. With a 70:30 feed ratio in VDF/MAF-TBE (Entries #6, 8 and 10), the alternating structure of the VDF/MAF-TBE is sustained, and the 1:1 ratio in the copolymer is not disturbed by the introduction of PFP, as evidenced by the presence of the signal at −94 ppm. Thus, the microstructures of the terpolymers consists in alternating poly(VDF-*alt*-MAF-TBE) micro-blocks randomly separated by PFP units (i.e., ~[(MAF-TBE-VDF)_n_-PFP]_m_~). This is consistent with the poly[(VDF-*alt*-MAF)-*co*-HFP] terpolymer previously reported because of the non-propagation of HFP [25,27,58]. When the VDF feed percentage increases, the proportion of alternating poly(VDF-*alt*-MAF-TBE) dyads decreases while that of PVDF micro-blocks increases. This was evidenced from the ^19^F NMR (Figure 4), where the intensities of PVDF signals centered at −91, −113 and −116 ppm, characteristic of the PVDF micro-blocks, strongly increased. The third monomer amount is crucial, as evidenced from Entries #6 to #10 in Table 1. In the case where VDF was introduced in strong excess compared to MAF-TBE (Entry #9), the yield of the obtained terpolymer was poor. Thus, the MAF-TBE incorporation being lower as a 90:10 VDF/MAF-TBE feed ratio does not lead to any poly(VDF-*alt*-MAF-TBE) (as mentioned earlier [25] for the copolymerization of VDF with MAF). However, with more MAF-TBE in the feed (Entry #10), both the yield and the PFP incorporation in the terpolymer increased.

However, as noted above, a high initial PFP concentration seems to inhibit the terpolymerization reaction as a confirmation of the poor reactivity of PFP: for example, 50.0 and 68.7 mol% of PFP in the feed led to only 5.1 and 4.5 mol% of PFP in the terpolymer with 20 and 16% yields, respectively (Entries #8 and #6 in Table 1).

Signals ranging between −68.1 and −70.6 ppm, and between −61 and −65 ppm, were assigned to the trifluoromethyl groups of MAF-TBE and PFP units, respectively, enabling the assessment of the composition of the terpolymers (Table 1) according to the following equations:$$\%VDF=\frac{I_{VDF}}{I_{VDF}+I_{PFP}+I_{MAF-TBE}}\times 100$$
$$\%PFP=\frac{I_{PFP}}{I_{VDF}+I_{PFP}+I_{MAF-TBE}}\times 100$$
$$\%MAF-TBE=\frac{I_{MAF-TBE}}{I_{VDF}+I_{PFP}+I_{MAF-TBE}}\times 100$$
where:$$I_{VDF}=\frac{\left(I_{-91}+I_{-94}+I_{-110}+I_{-113}+I_{-116}\right)}{2},I_{PFP}=\frac{\left(I_{-61.2}+I_{-65}\right)}{3},\;I_{MAF-TBE}=\frac{\left(I_{-68}+I_{-70.6}\right)}{3}$$
and where *I_j_* represents the integral of the signal located at *j* ppm.

As already mentioned, MAF-TBE does not propagate [41], and when an alternating poly(VDF-*alt*-MAF-TBE) structure was produced, no reverse addition could be seen, and no ~CH_2_CF_2_-CF_2_CH_2_~ structure was observed compared to ~CH_2_C*(CF_3_)(CO_2_*t*Bu)CF_2_C*H(CF_3_)~. Both the fluorine atoms located between the two asymmetric carbons led to a very complex structure located in the −65 to −71 ppm range. The CF_2_ signal of the VDF-PFP dyad (~CH_2_C***F***_2_-CF_2_CH(CF_3_)~), normally located in the −100 to −125 ppm, could not be noted. From these observations, a mechanism for this radical terpolymerization can be proposed (Figure 3)**.**

The decomposition of both Trigonox^®^ 101 and DTBP could first generate a tBuO^•^ radical that further rearranges into methyl radical upon heating [59,60]. These R^•^ radicals resulting from the decomposition of the initiators react onto VDF, generating a RCH_2_CF_2_^•^ that displays a very electrophilic character, thus preventing any interaction with the PFP monomer. Then, this radical can react with VDF if in excess (>75 mol%) or with MAF-TBE, which drives the macroradical towards an alternating structure. The macroradical can thus propagate according to Figure 3, taking into account that PFP also does not propagate.

If the yield is low (<20 mol%), macroradical (14) (Figure 4) does not occur. It is worth noting that the radical (co)polymerization of fluorinated monomers is usually terminated by recombination (and not disproportionation) [1]. Thus, the recombination of two growing macroradicals bearing PFP and VDF final units can be observed in Figure 4.

The -CF_2_- group between both asymmetric carbons is different from that adjacent to the CH_A_/H_B_ of VDF that gives a signal at −94 ppm (~CH_2_C(CF_3_)(CO_2_tBu)-**CF_2_**CH(CF_3_)-CH_2_CF_2_~). Thus, the signal at −95 ppm was assigned to the CF_2_ group (Figure 4, peak **e**).

Because of the non-regioselectivity, the addition of VDF could be normal (-CH_2_-CF_2_) or reverse (–CF_2_-CH_2_-), leading to the following possible structure: ~CH_2_CF_2_-CH_2_C(CF_3_)(CO_2_tBu)-CH_2_CF_2_-CH_2_C(CF_3_)(CO_2_t-Bu)-CF_2_CH(CF_3_)~. This structure is at the origin of the multiplicity of the signal observed at −68.1 ppm. Consequently, the micro-structure of the terpolymers synthesized with a PFP feed proportion ranging from 25 to 50 mol% and with an alternating VDF-MAF-TBE dyad is ~([CH_2_C(CF_3_)(CO_2_tBu)-CH_2_CF_2_]_n_-CF_2_CH(CF_3_))_p_~.

The ^1^H NMR spectrum of the terpolymer based on PFP, VDF and MAF-TBE (Appendix A) exhibits the characteristic signals of VDF (methylene normal additions at 2.9 ppm, while reverse additions at 2.2 ppm could not be detected), and that of MAF-TBE (*tert*-butyl at 1.5 ppm and methylene at about 1.9 ppm), while the signals characteristic of the PFP unit are not observed (should be in the 3.6–5.2 ppm range). The peak at 1.0 ppm was assigned to the methyl groups of the initiator.

All copolymers and terpolymers were purified and characterized, and their thermal properties were studied (Table 3, Appendix A in the Supporting Information). The assessment of number-average molar masses (M_n_) (Table 3, Appendix A) shows that high VDF content in the feed increases the molar masses in contrast to a PFP increase, which decreases them. The highest molar masses were reached when the feed VDF amount was high and that of PFP was low. PDI were often below 1.5, consistent with a low M_n_.

The thermal properties of the co- and terpolymers were assessed by means of TGA and DSC. With the exception of the poly(PFP-*co*-VDF) copolymers with a glass transition temperature of −29 °C, all the others exhibit a positive T_g_ within the 40 to 60 °C range, which can be linked to the presence of bulky trifluoromethyl and *tert*-butyl side groups. The introduction of PFP units in the PVDF chains might disturb the organization of PVDF units due to the presence of trifluoromethyl side groups that lower the intramolecular interactions between chains [20,22,23]. The introduction of MAF-TBE units has a much more profound influence on the chain organization due to the presence of the bulky side-groups (*tert*-Bu), leading to a T_g_ above room temperature. Table 3 displays the decomposition temperatures at 10 wt% loss obtained by TGA (Appendix A) in nitrogen and air. It can be seen, particularly with the copolymers from Entry #2, that the higher the MAF-TBE content, the lower the thermostability due to the elimination of 2-methylpropene (or isobutylene) followed by a decarboxylation reaction induced by the MAF-TBE units upon heating [42,44,45,61] (Figure 5). No significant difference was observed between the thermal analyses conducted under nitrogen and air, indicating that the decomposition of such copolymers was not further influenced by the oxidation.

#### 2.2.2. Radical Terpolymerization of PFP with VDF and HFP

To improve the reaction yield and to favor the termonomer-induced copolymerization conditions, a conventional radical terpolymerization of PFP with VDF and HFP was investigated (Figure 6). Fair to good yields (30–60%) were obtained (Table 2 Entries #4–6), while the yield was inversely proportional to the PFP fraction in the feed.

In the ^19^F NMR spectrum of the poly(PFP-*ter*-VDF-*ter*-HFP) terpolymer (Figure 5), several types of signals were found and assigned to each monomer units: −60; −65 ppm [-CF_2_CH(CF_3_)-]; −71; −77.5 ppm. [-CF_2_CF(CF_3_)-]; −91; −94 ppm (-CH_2_-CF_2_-CH_2_-CF_2_-head to tail chaining); −108; −113; −116 ppm (-CH_2_-CF_2_-CF_2_-CH_2_-); −110 ppm [-CF_2_CF(CF_3_)- CH_2_-CF_2_-CF_2_-CF(CF_3_)] and –119 ppm [-CF_2_CH(CF_3_)-]. The peaks located between −95 ppm and −103 ppm could be attributed to a CF_2_ group in VDF between an HPF and a PFP unit, –CF_2_CF(CF_3_)CF_2_CH_2_CF_2_CF(CF_3_)-. As above, integrals of characteristic signals enabled us to determine the composition of the terpolymer.

The ^1^H NMR spectrum (Appendix A in ESI) of poly(PFP-*ter*-VDF-*ter*-HFP) terpolymer exhibits normal VDF addition (2.8 and 3.6 ppm) but the absence of reverse VDF additions (2.4 ppm). The methyne protons of PFP are located between 4.2 and 4.7 ppm (not shown in Appendix A). Other peaks include the methyl group from the initiator at 1.2 and 1.8 ppm resulting from **CH_3_**-CH_2_-CF_2_~ (^3^J_HH_ = 7 Hz) or **CH_3_**-CF_2_-CH_2_~ (triplet, ^3^J_HF_ = 15 Hz), respectively [20], as the *tert*-butoxy radical released from the initiator rearranges into methyl radicals (and acetone) [56,57], which further initiate the terpolymerization.

Molar masses were estimated from the number of PFP, HFP and VDF units in the terpolymer from the integral ratio between the normal and reverse VDF signals compared to that of the methyl end groups from the initiator. Practically, this means:$$\bar{M_{n}}=N_{VDF}\times M_{VDF}+\alpha_{1}N_{VDF}\times M_{HFP}+\alpha_{2}N_{VDF}\times M_{PFP}+M_{2CH^{•}{}_{3}}$$
where *MVDF* = 64 g·mol^−1^*, MHFP* =150 g·mol^−1^ and *MPFP* = 132 g·mol^−1^.

Additionally, NVDF=I2.3−2.5+I2.8−3.52I1.2+I1.86, number of VDF units in the polymer chain,
$$\alpha_{1}=\frac{\%molHFP}{\%molVDF}\;\mathrm{and}\;\alpha_{2}=\frac{\%molPFP}{\%molVDF}$$

As in the above terpolymers, results from Table 2 indicate that the introduction of PFP in the polymer lowers the molar masses as, for example, that of poly(VDF-*co*-HFP) copolymer is ca. 40,000 g·mol^−1^ (eq. PMMA), while that of the terpolymer barely reaches 10,000 g·mol^−1^ even in the absence of any transfer.

The thermal properties of the co- and terpolymers were also assessed by DSC and TGA (Table 2 and Appendix A in ESI). All samples have a T_g_ value below −28 °C and no melting point, indicating they are all amorphous. As expected, the introduction of HFP and/or PFP disturbs the regular crystalline structure of PVDF, resulting in a T_g_ (>−40 °C). All polymers have a 10% weight loss decomposition temperature (T_d,10%_) above 230 °C, and are thermostable except for poly(PFP-*co*-HFP), for which it is only 109 °C due to its low molar mass (evaporation instead of decomposition upon heating). However, the thermal stability of these co- and terpolymer remains lower than that of poly(VDF-*co*-HFP), which might explain why the Montedison Company gave up their research on PFP in the 1970s to attempt competing with poly(VDF-*co*-HFP) elastomers.

#### 2.2.3. Radical Terpolymerization of PFP with VDF and PMVE

Similar initiators were used as in above terpolymerizations and the results are listed in Table 2. After purification, the resulting terpolymers were characterized by NMR spectroscopy.

As noted above, besides the characteristic signals assigned to VDF units (−91 to −94 ppm) and reverse additions (−108 to −116 ppm), the ^19^F NMR spectrum (Figure 6) displays the CF_2_ signals of VDF in VDF-PMVE dyads at −111 and −122 ppm and those observed at −111 ppm for poly(PFP-*co*-VDF) copolymers (see Section 2.1). As observed in previous work [57], in contrast to quite poor yields obtained from copolymerization, the terpolymerization confirms the tendency to produce terpolymers in greater yields and containing a higher PFP amount (Table 2), as can be highlighted from NMR spectra. Actually, the ^19^F NMR spectrum (Figure 6) exhibits the characteristic signals of PMVE (singlet centered at −52 ppm assigned to OCF_3_, and at −145 and −122 ppm for CF group) [27,30] and of PFP (CF_3_ at −60 and −65 ppm and a complex AB system at −117 and −124 ppm, corresponding to the CF_2_ group flanked between two asymmetric carbon atoms). The ^1^H NMR spectrum (Appendix A in ESI) clearly shows the characteristic signals of the methyne protons of PFP at 3.3–3.6 ppm and traces of methyl groups at 1.1 ppm resulting from the end group of the initiator from the direct initiation. The spectrum also exhibits the characteristic signals of the methylene groups of VDF (normal additions) with a broad complex multiplet centered between 2.8 and 3.6 ppm.

#### 2.2.4. Radical Terpolymerization of PFP with VDF and TFP

Radical terpolymerizations of PFP, VDF and TFP were initiated by Trigonox^®^ 101, and the results are listed in Table 2. ^19^F and ^1^H NMR spectra are displayed in Appendix A, respectively.

Although the ^1^H NMR signals of the respective PFP, TFP and VDF units were just described, in the present case, the methylene protons of both VDF and TFP overlap in the 2.3–2.6 ppm region (Appendix A). The multiplet in the 2.8–3.2 ppm region was assigned to the methyne protons of TFP and PFP overlapping with the methylene protons of VDF.

Besides the characteristic signals of VDF units in the −91 to −94 ppm range, −114 and −116 ppm, the ^19^F NMR spectrum (Appendix A) displays the signals centered at −60 and −66 ppm assigned to the trifluoromethyl groups of PFP, while that of TFP is present at −71 ppm [31,33,34]. The signal centered at −95 ppm is assigned to the VDF-TFP dyads as a reverse tail-to-head additions. This tends to confirm that PFP does not propagate nor could cross propagate with TFP because of the high reactivity of VDF towards both TFP and PFP. This means that a VDF unit is always inserted after a TFP or PFP unit. As above, the introduction of VDF in the polymerization induced a strong increase in the molar masses as can be seen from the SEC results, which can be explained by a termonomer-induced copolymerization (TIC), as described above [38].

From the study of the thermal properties, only amorphous fluorinated terpolymers were obtained, the T_g_ values of these co- and terpolymers ranging from −53 °C to −28 °C, and no melting points were observed in the DSC thermograms. Additionally, the thermal stability of the terpolymers was better than those of the copolymers, as a result of their higher molar masses. Finally, as for all above cases, no significant differences were noted for the decomposition of these fluoropolymers under nitrogen or air, indicating a stability towards oxidation.

#### 2.2.5. Radical Terpolymerization of PFP with CTFE and M_3_ Monomer

To provide insights on the high reactivity of CTFE with 3-isopropenyl-α,α-dimethylbenzyl isocyanate (m-TMI) [37] and vinyl ethers [35], which could increase the PFP reactivity, terpolymerizations of these monomers with PFP were also attempted.

The same previous strategies were applied to CTFE and PFP with three non-fluorinated comonomers: vinylene carbonate (VCA), ethyl vinyl ether (EVE) and m-TMI. All three of them yield alternating copolymers with CTFE [35,62,63,64]. Polymerizations were carried out according to Figure 7 (initiated by Trigonox^®^ 101 in C_4_F_5_H_5_). Results are summarized in Table 4.

#### 2.2.6. Radical Terpolymerization of PFP with CTFE and m-TMI

From all terpolymerizations of PFP with CTFE and a third monomer, the one involving m-TMI produced the best results: the yield was 30 wt% and the PFP content in the terpolymer was about 35 mol%. This is in agreement with the previous results, as the yield for the copolymerization of CTFE with m-TMI was 50 wt%, and it decreased when PFP was introduced.

As expected, in the ^19^F NMR spectrum of the poly(PFP-*ter*-CTFE-*ter*-mTMI) terpolymer (Appendix A), the characteristic -CF_3_ signals from PFP can be found at −60 ppm. On the other hand, the presence of CTFE units is supported by the signals centered at ca. −105 ppm which can be assigned to the two non-equivalent fluorine of -CF_2_- (F_A_ and F_B_), forming the AB part of an ABX system. Other signals from CTFE (-CF-) can be found at −95, −120, −130 and −153 ppm [35,62,63]. The signal at −80 ppm could be assigned to trifluoromethyl of PFP adjacent to TMI.

The ^1^H NMR spectrum of poly(PFP-*ter*-CTFE-*ter*-mTMI) terpolymer (Appendix A) displays a complex signal in the 3.2–3.6 ppm range assigned to the proton of the methyne group of PFP (the broad signal arises from the presence of the asymmetric carbon). The aromatic protons of m-TMI are located at in the 7.5–8.0 ppm range. The methyl protons either on the backbone or in the side group are located at 2.4 and 1.0 ppm, respectively.

#### 2.2.7. Radical Terpolymerization of PFP with CTFE and VDF

Poly(VDF-*co*-CTFE) copolymers are commercially available [65], but the incorporation of PFP in the terpolymer was extremely low (<1%) and the yield was about 24 wt%. However, VDF and CTFE are reactive towards each other (r_CTFE_ = 0.75 and r_VDF_ = 0.73 at 80 °C [66,67]), and the composition indicates a similar proportion of CTFE and VDF, which could suggest that the structure of the terpolymer is mostly an alternating structure of CTFE and VDF, randomly separated by a PFP [57] unit. However, VDF and CTFE do not usually form alternating copolymers [35]; it is thus suggested that the apparent alternating composition is just fortuitous and that the nature of the terpolymer is essentially random but in equal proportion, with a PFP unit inserted from time to time. The NMR analyses tend to confirm this random structure, as evidenced by the presence of micro-blocks.

In the ^19^F NMR spectrum of the poly(PFP-*ter*-CTFE-*ter*-VDF) terpolymer (Appendix A), and from the previous sections, the characteristic signals from PFP can be found at −60 ppm (CF_3_) and −120 ppm (-CF_2_-), while that of CTFE are located at −108 ppm (–CF_2_- (F_A_ and F_B_)) and at −92, −120, −127 and −153 ppm (-CF-) [35,39]. Signals of VDF are centered in the −91 to −95 ppm area but also at −113 and −116 ppm (-CF_2_-). It is also noted that the peaks on this spectrum are very narrow, coherent with a low-molar-mass polymer, as indicated by SEC analyses (990 g mol^−1^).

The ^1^H NMR spectrum (Appendix A) shows signals in the 3.0 to 3.5 ppm range that are assigned to the methyne proton of the VDF normal additions overlapping with the signal of the methyne protons borne by the asymmetric carbon of PFP. VDF reverse additions are noted at 2.4 ppm, while the signals at 0.9 and 1.2 ppm are assigned to the methyl protons of the initiator.

#### 2.2.8. Radical Terpolymerization of PFP with CTFE and VCA

A previous study reports that copolymers based on CTFE and vinylene carbonate (VCA) are alternated [65], but in the present case, poly(CTFE) micro-blocks were formed with a random distribution with PFP and VCA units in a 22 mol% yield.

As mentioned in the sections above, the characteristic signals of the three monomers could be found on the ^19^F NMR spectrum (Figure 7): PFP at −60 and −65 ppm (CF_3_-) and −120 ppm (-CF_2_-); CTFE at −108 ppm (–CF_2_- F_A_ and F_B_), −120, −130 and −153 ppm (-CF-).

The ^1^H NMR spectrum (Appendix A) also exhibits the characteristic signals of PFP (at 2.9 ppm, -CH-) and those of the methyne groups of VCA in the backbone (-CH-CH-) located at 6.0–6.5 ppm. The signals at 1.2 and 1.5 ppm are assigned to the end groups resulting from the initiator.

#### 2.2.9. Radical Terpolymerization of PFP with CTFE and EVE

Ethyl vinyl ether (EVE) was selected since it leads to perfectly alternated poly(CTFE-*alt*-EVE) copolymers [62,63,68] according to the acceptor–donor polymerization principle [1,36,64]. Similar results as for the terpolymerization in the presence of VCA were obtained, although VCA is known to be less reactive than EVE towards its copolymerization with CTFE [65]. However, the obtained poly(PFP-*ter*-CTFE-*ter*-VDF) terpolymer contained a low PFP amount (1.6 mol%) and it can be suggested that both alternating oligo(CTFE-*alt*-EVE) structure [62,63,68] and oligo(CTFE) micro-blocks were found (ratio 79:19 mol%), as evidenced by the signal at −127 ppm attributed to CTFE-CTFE dyads [39].

As mentioned in the previous sections, the characteristic signals of the three monomers could be found on the ^19^F NMR spectrum (Appendix A): PFP at −60 and −65 ppm (CF_3_) and −105 ppm (-CF_2_-); CTFE at −108 and −114 ppm (–CF_2_- F_A_ and F_B_), −121 (alternated structure), −127 (microblock of PCTFE), −134 and −140 ppm (-CF-).

Appendix A exhibits the ^1^H NMR spectrum of poly(PFP-*ter*-CTFE-*ter*-EVE) terpolymer. The signals at 3.8 and 1.2 ppm were assigned to the methylene and methyl groups in the ethoxy side group (-OC_2_H_5_) of EVE unit and that of the methylene and methyne protons (backbone) of the EVE are evidenced by the broad peaks centered at 2.9 and 4.7 ppm, respectively. The signal of the methyne proton of PFP is overlapping with that of the methylene of EVE. The signal at 1.5 ppm results from the initiator [62] while the signal at 6.2 ppm is assigned to the terminal proton CF_2_H resulting from a chain transfer reaction [40].

Molar masses (Table 4) vary significantly from 1000 (with VDF as third monomer) to 37,000 g·mol^−1^ (with EVE as a third monomer) with a polymodal distribution, as reflected by a large dispersity. As in previous cases, the PFP incorporation in the polymer chain has a strong influence on the reaction yield, the molar masses and the polydispersity.

### 2.3. Thermal Properties of Co- and Terpolymers

A study of the thermal properties of the co- and terpolymers (Table 4) showed that PFP incorporation decreased the T_g_ in the CTFE copolymers from 52 °C for pure PCTFE [35] to about 8 °C, independently from the PFP amount, although small, in the copolymer. This decrease could be in part due to the low molar masses but tends to indicate that the introduction of PFP softens the co- and terpolymers.

The poly(PFP-*ter*-CTFE-*ter*-VCA) terpolymer displays a lower T_g_ than that of poly(CTFE-*co*-VCA) (49 °C vs. 70–120 °C [65]), which could be assigned to the presence of PFP (i.e., CF_3_ lateral group) and the low molar masses of the terpolymers (Appendix A). In the case of poly(PFP-*ter*-CTFE-*ter*-mTMI) terpolymer, the observed T_g_ is also low even if the aromatic ring of m-TMI is supposed to increase it. However, as for the poly(CTFE-*alt*-mTMI), molar masses are low, and it can be argued that the steric hindrance of this compound could explain the low T_g_, as the presence of a -C_6_H_4_-C(CH_3_)_2_-N=C=O lateral group decreases the intermolecular interactions.

Because there is no significant difference between the decomposition temperature obtained in nitrogen or air, these fluorinated copolymers are stable towards oxidation. The introduction of VCA or EVE units has a similar influence upon the thermal stability; in contrast to m-TMI, which induces a lower decomposition temperature because of the low molar masses (as for copolymers).

In the case of the co- and terpolymers of PFP, CTFE and M_3_ (VDF, mTMI, VCA and EVE), most poly(CTFE-*co*-M_3_) copolymers have mainly an alternated microstructure, the introduction of PFP (due to its poor reactivity) is likely to decreasing the affinity towards alternating micro-structures, as highlighted from their NMR spectra. Depending of the composition and the type of third monomer (M_3_), the T_g_ of the terpolymers varies from −35 °C to +50 °C, and all polymers have a thermal stability above 200 °C (Appendix A).

## 3. Experimental Section

### Materials

All reagents were used as received unless stated otherwise.

1,1,3,3,3-pentafluoropropene, PFP, and 3,3,3-trifluoropropene, TFP, were kindly supplied by the Great Lakes Chemical Corporation (or Chemtura Company, West Lafayette, IN, USA). 1,1-Difluoroethylene (vinylidene fluoride, VDF), hexafluoropropylene (HFP) and 1,1,1,3,3-pentafluorobutane (Solkane^®^ 365 mfc) were kindly supplied by Arkema (Pierre-Bénite, France), Solvay S.A. (Tavaux, France) and Elf Atochem (now Arkema, Colombes, France), while chlorotrifluoroethylene (CTFE) was supplied by Honeywell (Morristown, NJ, USA). *Tert*-butyl 2-trifluoromethacrylate (MAF-TBE) and perfluoromethylvinyl ether (PMVE) were kindly supplied by Tosoh Finechemical Corporation (Shunan, Japan) and Dupont Performance Elastomers LLC (Wilmington, NC, USA), respectively. *Tert*-butylperoxypivalate in solution of isododecane (Trigonox^®^ 25-C75, TBPPi) (purity 75 wt%), 2,5-bis(*tert*-butylperoxy)-2,5-dimethylhexane (Trigonox^®^ 101) (purity 92 wt%) and di-*tert*-butyl peroxide (DTBP), and azobisisobutyronitrile (AIBN) were gifts from Akzo Nobel (now Nouryon, Chalons sur Marne, France) and Merck (Darmstadt, Germany), respectively. Calcium hydride (CaH_2_), sodium sulfate (Na_2_SO_4_), acetonitrile, vinylene carbonate (VCA), 3-isopropenyl-α,α-dimethylbenzyl isocyanate (mTMI) and all other chemicals were purchased from Sigma-Aldrich (Saint Quentin-Fallavier, France). Acetonitrile was freshly distilled over calcium hydride and degassed by bubbling argon before use. Deuterated solvents (chloroform, acetone, DMF) used for the NMR spectroscopy were purchased from Euroisotop (Grenoble, France) (purity > 99.8%).

## 4. Characterization

### 4.1. NMR

The NMR spectra were recorded on Bruker AC 400 instruments (Bruker, Billerica, MA, USA), using deuterated chloroform or acetone as the solvent and tetramethylsilane (TMS) (or CFCl_3_) as the references for ^1^H (or ^19^F) nuclei. Coupling constants and chemical shifts are in hertz (Hz) and part per million (ppm), respectively. The experimental conditions for recording ^1^H [or ^19^F] NMR spectra were as follows: flip angle 90° [or 30°], acquisition time 4.5 s [or 0.7 s], pulse delay 2 s [or 5 s], number of scans 36 [or 64] and a pulse width of 5 μs for ^19^F NMR.

### 4.2. SEC

Size-exclusion chromatography (SEC) was carried out in dimethyl formamide (DMF) containing LiBr (5%) at 30 °C at a flow rate of 0.8 mL/min by means of a Spectra Physics Winner Station, a Waters Associate R 401 differential refractometer and a set of two columns PL Gel (Polymer Laboratories, now Agilent, Santa Clara, CA, USA) 5 m 100 Å connected in series. Monodispersed (PMMA) standards were used for calibration (100–100,000 g·mol^−1^) as conventional polymer standards since no fluorinated ones are commercially available. Aliquots were sampled from the reaction medium, diluted with DMF up to a known concentration (C_p,t_) ca. 5 wt%, filtered through a 0.45 µm PTFE Chromafil Membrane and finally analyzed by GPC under the conditions described above.

### 4.3. FTIR

IR spectra were recorded on a Nicolet 510P Fourier Transform Infrared (FTIR) spectrometer from KBr pellets (10 wt%), and the intensities of the absorption bands (cm^−1^) were labeled strong (s), medium (m) or weak (w). The accuracy was ±2 cm^−1^.

### 4.4. TGA

Thermogravimetric analyses (TGA) were performed with a TGA 51 apparatus from TA Instruments, in air and nitrogen, at the heating rate of 5 °C·min^−1^ from room temperature up to a maximum of 580 °C. The sample sizes varied between 10 and 15 mg.

### 4.5. DSC

Differential scanning calorimetry (DSC) measurements were conducted using a Perkin Elmer Pyris 1 apparatus (Perkins Elmer, Waltham, MA, USA). Scans were recorded at a heating rate of 10 °C·min^−1^ from −100 to +100 °C (interval was calibrated with indium, T_m_ = 156.6 °C), and the cooling rate was 20 °C·min^−1^. A second scan was required for the assessment of the T_g_, defined as the inflection point in the heat capacity jump. The sample size was about 10 mg.

## 5. Synthesis

### 5.1. Autoclave

The radical copolymerizations of PFP with fluorinated and/or non-fluorinated comonomer(s) were performed in a 160 mL Hastelloy autoclave Parr system (HC 276) equipped with a manometer, a mechanical Hastelloy anchor stirrer, a rupture disk (bearing as much as 200 bars) and inlet and outlet valves. An electronic device regulated and controlled both the stirring and heating of the autoclave. Prior to reaction, the autoclave was pressurized with 30 bars of nitrogen for 1 h to check for leaks. The autoclave was then conditioned for the reaction with several nitrogen/vacuum cycles (10^−2^ mbar) to remove any trace of oxygen. The liquid and dissolved solid phases were introduced via a funnel tightly connected to the autoclave, and then the gases were transferred by double weighing (i.e., the difference in weight before and after filling the autoclave with the gases).

### 5.2. Radical Copolymerization of PFP

Copolymerizations were carried out for 20 h either in acetonitrile (CH_3_CN) or in 1,1,1,3,3-pentafluorobutane (C_4_H_5_F_5_, Solkane^®^ 365 mfc), both solvents being able to dissolve the fluorinated monomers without inducing any chain transfer reaction. These reactions were initiated by either Trigonox^®^ 101 at 134 °C and/or DTBP at 140 °C (half time of 1 h for both initiators). C_0_ represents the initiator (I)/monomers (M) initial molar ratio and was chosen in the 0.01–0.03 range (i.e., 1 to 3 mol% of initiator compared to total monomer content).
$$C_{0}=\frac{{\left[I\right]}_{0}}{\sum {\left[M\right]}_{0}}$$

## 6. Conclusions

PFP is a poorly reactive monomer that cannot homopolymerize, and we studied its radical copolymerization with six fluorinated comonomers (VDF, HFP, CTFE, MAF-TBE, TFP and PMVE). Though disappointing results were noted, their terpolymerization with other monomers that favored a termonomer-induced copolymerization (TIC) was attempted. From these results, a possible comparison in the reactivity can be suggested. The reactivity ranking of PFP in the copolymerization is in the following increasing order: MAF-TBE ≈ CTFE < HFP < TFP < VDF < PMVE. However, the ranking for the reaction yield is almost opposite: HFP < CTFE < PMVE < VDF < TFP < MAF-TBE. Furthermore, the decreasing yields for the terpolymers is in the following order: PFP/VDF/HFP > PFP/VDF/MAF-TBE > PFP/CTFE/VCA > PFP/VDF/PMVE = PFP/VDF/TFP. The decreasing order of reactivity of PFP in the terpolymers is as follows: m-TMI > VCA > PMVE > TFP ≈ HFP > MAF-TBE. From the terpolymerization, two types of products were obtained: fluorinated thermoplastics (either powder or viscous oils), with a positive glass transition temperature in the 40–50 °C range [poly(PFP-*ter*-VDF-*ter*-MAF-TBE) and poly(PFP-*ter*-CTFE-*ter*-VCA)], and the others were amorphous with a low T_g_ ranging from −53 °C to −30 °C (as viscous liquids) [poly(PFP-*ter*-VDF-*ter*-HFP), poly(PFP-*ter*-VDF-*ter*-PMVE) and poly(PFP-*ter*-VDF-*ter*-TFP]. All these products exhibit a fair to good thermal stability in air (T_d,10%_ > 190 °C) and are soluble in polar solvents (acetone, THF, C_4_F_5_H_5_, etc.), but have rather low molar masses (2000 ≤ Mn< 12,000 g·mol^−1^ (with PMMA standard)) with a dispersity ranging from 1.2 to 2.0. It was found that high VDF content in the feed increased the molar masses, while increasing the PFP content decreased it. The highest molar masses were reached when the VDF feed proportion was high and that of PFP was low. Further work is necessary involving electron-donating monomers such as ethylene, norbornene, *N*-vinyl pyrrolidone, paramethoxystyrene, other vinyl ethers, etc.

## Data Availability

Not applicable.

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
