# Peer review of "Are (Co)Polymers of 1,1,3,3,3-Pentafluoropropene Possible?"

_molecules, 2023, doi:10.3390/molecules28124618_

Round 1
Reviewer 1 Report
This manuscript represents a very detailed and exhaustive study by Ameduri and coworkers on the reactivity/polymerizability of the monomer 1,1,3,3,3-pentafluoropropene (PFP). Although not expected to be a very reactive monomer based on the limited amount of information in the literature, the authors attempt to answer the question(s) posed in the title of their manuscript. Interesting to this Reviewer were the observations were the presence of PFP as a comonomer facilitated in a sense the 'oligomerization' of monomers such as HFP and MAF-TBE that are well known to typically not homopolymerize.
This manuscript should be published after considerable revision, much of which is grammatical, typesetting, etc.
However, the following technical issues should be addressed:
1. Pg 1, Line 29: 1,1,3,3,3-pentafluoropropene is 2H-PFP, as the remaining hydrogen atom is on carbon atom 2. Likewise, 1,2,3,3,3-pentafluoropropene is 1H-PFP, as the remaining hydrogen atom is thus on carbon atom 1. Furthermore, this compound has two geometric isomers that were not mentioned. Thus, you also need to correct throughout as well as make sure that the cited references match the correct structural (and perhaps geometric) isomer.
2. An exhaustive literature search was not done, but it appears that the Introduction is missing primarily more recent patents in this area. For example, please see WO2022021187, US8969432, and WO 2002092687.
3. Likewise, Pg 2, line 64, “1H-PFP” should be “2H-PFP”.
4. Pg 2, line 105: Perhaps it should be pointed out that PMMA standards may not be that representative for fluoropolymers in SEC, even though this is well known to specialists in the field.
5. Either some data/Entries (Runs) are missing, or the Entries in Table 3 are mislabeled. Table 3 states on Lines 463-464 that the data is for co- and terpolymers of PFP with VDF and MAF-TBE, while on Line 529 and Lines 548-549 (Figure 6), it is mentioned that data for the terpolymer from PFP, VDF, and PMVE are include in Table 3. Also, on Line 522, an indication is given that data on the terpolymer from PFP, VDF, and TFP are also given in Table 3, and they are NOT. Please update Table 3 and/or text to make correct and consistent, including with the Supporting Information (SI).
From the amount of corrections being suggested below, one can only conclude that the current presentation is a mess. On the other hand, these authors normally submit manuscripts that are in much better shape, but perhaps it is time to be reminded that it is not the role of the Reviewers and Editors to clean-up such submissions. Please address the following issues:
The manuscript is replete with minor mistakes, so a very thorough proofreading and numerous corrections need to be made. A few examples include the following:
1. In the Title, please remove the extra spacing before the first and third “3”.
2. Be consistent, as you have used both “M3” and “MÚ3Ú” throughout the manuscript, including “athird” in the title for Table 2 (Line 249).
3. In the Abstract, “tert” needs to be in italics, like it is everywhere else in the manuscript.
4. Pg 1, Line 32: Remove the apostrophe in “1950-1960’s”. It should be “1950-1960s”. And, likewise on Pg 16, Line 526 for “1970’s”, which should be “1970s”.
5. Pg 1, Line 40, and Pg 12, Line 430: The reference call out should come before the punctuation.
6. The terms “wt.%”, “wt %”, and “wt%” all appear either in the text and/or Tables. The correct one should be consistently used throughout.
7. Pg 3, Lines 118-121: “Perkin Elmer” is mis-spelled, and the calibration to an indium standard is mentioned twice – redundant.
8. Pg 3, Line 127: “160 mL” should be “160-mL”.
9. Pg 3, Line 128: “anchor” - ? Is this a stirrer in the shape of an anchor? Please reword.
10. Pg 3, Line 142: “interval” - ? Do you mean “ratio” ? That is a ratio of 0.01-0.03 to 1.00 of initiator to monomer or do you mean concentration, although no units were given. Please clarify.
11. On Lines 150-151, 171, 176, 184, 207-208, 247, 275, 320, 377-378, 384-385, 395, 398, 410-411, 414, 428, 435, 439, 448-449, 466, 467-468, 480-481, 485-486, 495, 496, 500-501, 525, and 530, the following message appears where the seems to be either missing text and/or references – “Error! Reference source not found ..” The authors apparently did not double-check the built pdf file from their submissions very well, but this needs to be done now.
12. In Table 1, it would be best if the letters in terms like “solvent”, “PFP”, “CHÚ3ÚCN”, could appear all on the same line.
13. Lines 163 and 170: “<10 %” and “11 %” – Percent is a unitless number, so no space should be skipped between the number and the symbol “%”.
14. Lines 179, 229, 251, 254, 310, 316, 328, 363 (364 “m” missing in “ppm”), 426, 450-451, 502, 503, 531, 559, 560, 561, 570, 587, 600, 602, 613, 622, 638, 665, and 695: keep numbers and their units on the same line of running text, hyphens with their text, etc.
15. Line 186: Add a space in “Table1”. Also, please correct “Table2” and “Table4” in the SI.
16. Line 188: Compound sentence needs a comma after first “VDF”.
17. Line 195: Reword “… range [15-17] and while …” to “… range [15-17], while …”.
18. Line 278: Change “AS” to “As”.
19. Lines 290-291: Place figure caption on the same page as the figure is on.
20. Lines 300 and 382-383: Put “Termonomer Induced Copolymerization” in lower case.
21. You are inconsistent with the capitalization of “entry”/”Entry” and “entries”/”Entries”. The same is true for the term “run”/”Run”. Please correct and be consistent.
22. Line 351: Insert “such” between “copolymers” and “as”.
23. Improve line spacing after Line 420.
24. Line 551-552: Compound sentence – please add a comma after “101”.
25. The type in Schemes 3 and 4 is NOT legible, i.e., NOT dark enough when printed out, so the line thickness must be more narrow than other type.
26. Line 609: The SI gives 0.9 ppm rather than the 1.0 ppm point out in the text of the manuscript. Please make Figure correct and consistent.
27. Line 652: Insert space between “CTFE” and “[63]”.
28. Major problems exist with the figures in the SI file. For example, the manuscript and the Table of Contents (ToC) of the SI say that the 19F NMR and 1H NMR spectra of poly(VDF-co-MAFTBE) are Figures S11 and S12, respectively. However, when one looks for the actual spectra in the SI, one finds the 19F NMR spectrum of poly(VDF-co-MAFTBE) as Figure S15 and the 1H NMR spectrum of poly(VDF-co-MAFTBE) as Figure S6. ????? The 1H NMR spectrum of poly(PFP-ter-VDF-ter-HFP) is also totally messed up with respect to its Figure S## in the SI. In the manuscript and ToC of the SI, this is Figure S16; however, in the body of the SI, this is labeled as the second Figure S10 and discuss there as Figure S17. There is even a figure that is labeled “Figure S174”. I will not attempt to clean-up this mess, but please do so in your revision.
29. Line 671: Please change “Of” to “of”.
30. Line 672: Reword so you do not have passive and active verbs in the same sentence: “A study of the thermal properties of the co- and terpolymers (Table 4) showed …”.
31. You have used a fairly equal mixture of “ter-polymer” and “terpolymer” as well as “ter-polymerization” and “terpolymerization” throughout the manuscript. Please correct and be consistent.
32. Line 721: Please italicize the “N” in “N-vinyl”.
Author Response
I thank you for your email of May 24th 2023 related to our manuscript entitled " Are (co)polymers of 1,1,3,3,3-pentafluoropropene possible?" (Manuscript ID: molecules-2398580) which included the reviewers’ comments. I appreciate their constructive remarks.
I have carefully considered all the reviewers’ remarks and suggestions and revised the manuscript accordingly. The corrected or added parts are indicated by using the yellow highlight in the revised manuscript.
Actually, a main issue was the pdf conversion that was badly done (I requested that supplied to reviewers) and though we checked the submitted pdf file, it seems that another one was supplied to reviewers (that explained the “Error! Reference source not found ..” and many other issues).
Please find below our point-by-point response to the reviewers’ comments:
Reviewer 1
This manuscript represents a very detailed and exhaustive study by Ameduri and coworkers on the reactivity/polymerizability of the monomer 1,1,3,3,3-pentafluoropropene (PFP). Although not expected to be a very reactive monomer based on the limited amount of information in the literature, the authors attempt to answer the question(s) posed in the title of their manuscript. Interesting to this Reviewer were the observations were the presence of PFP as a comonomer facilitated in a sense the 'oligomerization' of monomers such as HFP and MAF-TBE that are well known to typically not homopolymerize.
This manuscript should be published after considerable revision, much of which is grammatical, typesetting, etc.
However, the following technical issues should be addressed:
- Pg 1, Line 29: 1,1,3,3,3-pentafluoropropene is 2H-PFP, as the remaining hydrogen atom is on carbon atom 2. Likewise, 1,2,3,3,3-pentafluoropropene is 1H-PFP, as the remaining hydrogen atom is thus on carbon atom 1. Furthermore, this compound has two geometric isomers that were not mentioned. Thus, you also need to correct throughout as well as make sure that the cited references match the correct structural (and perhaps geometric) isomer.
Answer: We have found the term « geometric isomer » unclear since the monomer we used (2H-PFP) does not have any other isomer ; in contrast to the other one (that can be E or Z). The difference of both the isomers (thanks for raising this point) has been corrected troughout the revised manuscript.
- An exhaustive literature search was not done, but it appears that the Introduction is missing primarily more recent patents in this area. For example, please see WO2022021187, US8969432, and WO 2002092687.
Answer: We appreciate to get these additional patents, two of them have been inserted in the introduction (the last one WO2022021187A1 being out of the scope and even does not claim any « fluor »).
- Likewise, Pg 2, line 64, “1H-PFP” should be “2H-PFP”.
Answer: Yes and this mistake has been corrected.
- Pg 2, line 105: Perhaps it should be pointed out that PMMA standards may not be that representative for fluoropolymers in SEC, even though this is well known to specialists in the field.
Answer: we have completed the sentence on this point.
- Either some data/Entries (Runs) are missing, or the Entries in Table 3 are mislabeled. Table 3 states on Lines 463-464 that the data is for co- and terpolymers of PFP with VDF and MAF-TBE, while on Line 529 and Lines 548-549 (Figure 6), it is mentioned that data for the terpolymer from PFP, VDF, and PMVE are include in Table 3. Also, on Line 522, an indication is given that data on the terpolymer from PFP, VDF, and TFP are also given in Table 3, and they are NOT. Please update Table 3 and/or text to make correct and consistent, including with the Supporting Information (SI).
Answer: we are sorry for the typo and have corrected « Table 3 » by « Table 2 » for the appropriate terpolymers.
Comments on the Quality of English Language
From the amount of corrections being suggested below, one can only conclude that the current presentation is a mess. On the other hand, these authors normally submit manuscripts that are in much better shape, but perhaps it is time to be reminded that it is not the role of the Reviewers and Editors to clean-up such submissions. Please address the following issues:
The manuscript is replete with minor mistakes, so a very thorough proofreading and numerous corrections need to be made.
Answer: as mentioned above, the pdf conversion was badly done and though we checked the pdf file, it seems that another one was supplied to the reviewers (that explained the “Error! Reference source not found” and issue of lining..)
A few examples include the following:
- In the Title, please remove the extra spacing before the first and third “3”.
Answer: this was OK in our word file and the conversion into pdf led to that unexpected mistake.
- Be consistent, as you have used both “M3” and “MÚ3Ú” throughout the manuscript, including “athird” in the title for Table 2 (Line 249).
Answer: corrected I caption of Table 2.
- In the Abstract, “tert” needs to be in italics, like it is everywhere else in the manuscript.
Answer: I systematically use « co » and « ter » in italics, but, again, the conversion in pdf did not take this into account.
- Pg 1, Line 32: Remove the apostrophe in “1950-1960’s”. It should be “1950-1960s”. And, likewise on Pg 16, Line 526 for “1970’s”, which should be “1970s”.
Answer: though we see both in articles, we have corrected.
- Pg 1, Line 40, and Pg 12, Line 430: The reference call out should come before the punctuation.
Answer: done.
- The terms “wt.%”, “wt %”, and “wt%” all appear either in the text and/or Tables. The correct one should be consistently used throughout.
Answer: we have corrected this and have kept them identical in the manuscript.
- Pg 3, Lines 118-121: “Perkin Elmer” is mis-spelled, and the calibration to an indium standard is mentioned twice – redundant.
Answer: yes and sorry for that redundancy.
- Pg 3, Line 127: “160 mL” should be “160-mL”.
Answer: corrected.
- Pg 3, Line 128: “anchor” - ? Is this a stirrer in the shape of an anchor? Please reword.
Answer: yes, the stirrer in the autoclave has the shape of an anchor.
- Pg 3, Line 142: “interval” - ? Do you mean “ratio” ? That is a ratio of 0.01-0.03 to 1.00 of initiator to monomer or do you mean concentration, although no units were given. Please clarify.
Answer: this is a range of molar ratio.
- On Lines 150-151, 171, 176, 184, 207-208, 247, 275, 320, 377-378, 384-385, 395, 398, 410-411, 414, 428, 435, 439, 448-449, 466, 467-468, 480-481, 485-486, 495, 496, 500-501, 525, and 530, the following message appears where the seems to be either missing text and/or references – “Error! Reference source not found ..”The authors apparently did not double-check the built pdf file from their submissions very well, but this needs to be done now.
Answer: we checked the pdf file that sound correct, in contrast to that sent to reviewers.
- In Table 1, it would be best if the letters in terms like “solvent”, “PFP”, “CHÚ3ÚCN”, could appear all on the same line.
Answer: Same problem of pdf file: the layout was suitable in the word file prior to the pdf conversion.
- Lines 163 and 170: “<10 %” and “11 %” – Percent is a unitless number, so no space should be skipped between the number and the symbol “%”.
Answer: OK, they were corrected.
- Lines 179, 229, 251, 254, 310, 316, 328, 363 (364 “m” missing in “ppm”), 426, 450-451, 502, 503, 531, 559, 560, 561, 570, 587, 600, 602, 613, 622, 638, 665, and 695: keep numbers and their units on the same line of running text, hyphens with their text, etc..
Answer: Same problem of pdf file.
- Line 186: Add a space in “Table1”. Also, please correct “Table2” and “Table4” in the SI.
Answer: probably same issue with pdf conversion since our word file was suitable.
- Line 188: Compound sentence needs a comma after first “VDF”.
Answer: done.
- Line 195: Reword “… range [15-17] and while …” to “… range [15-17], while …”.
Answer: the sentence has been cut into two parts and improved.
- Line 278: Change “AS” to “As”.
Answer: done
- Lines 290-291: Place figure caption on the same page as the figure is on.
Answer: same issue with pdf conversion.
- Lines 300 and 382-383: Put “Termonomer Induced Copolymerization” in lower case.
Answer: corrected.
- You are inconsistent with the capitalization of “entry”/”Entry” and “entries”/”Entries”. The same is true for the term “run”/”Run”. Please correct and be consistent.
Answer: this has been corrected. « Entry » throughout the manuscript.
- Line 351: Insert “such” between “copolymers” and “as”.
Answer: yes, done.
- Improve line spacing after Line 420.
Answer: done (pdf conversion)
- Line 551-552: Compound sentence – please add a comma after “101”.
Answer: done.
- The type in Schemes 3 and 4 is NOT legible, i.e., NOT dark enough when printed out, so the line thickness must be more narrow than other type.
Answer: this clear arrows also arise from the pdf conversion while the word file was correct.
- Line 609: The SI gives 0.9 ppm rather than the 1.0 ppm point out in the text of the manuscript. Please make Figure correct and consistent.
Answer: Figure S21 is an extension of the 1.0 to 9.0 ppm zone and does not show the presence of signal at 0.9 ppm. Figure S23 displays the signal at 0.9 ppm.
- Line 652: Insert space between “CTFE” and “[63]”.
Answer: done (the revised version states 65 instead)
- Major problems exist with the figures in the SI file. For example, the manuscript and the Table of Contents (ToC) of the SI say that the 19F NMR and 1H NMR spectra of poly(VDF-co-MAFTBE) are Figures S11 and S12, respectively. However, when one looks for the actual spectra in the SI, one finds the 19F NMR spectrum of poly(VDF-co-MAFTBE)as Figure S15 and the 1H NMR spectrum of poly(VDF-co-MAFTBE) as Figure S6. ?????
Answer: I am sorry for these confusions. The revised version has been corrected.
The 1H NMR spectrum of poly(PFP-ter-VDF-ter-HFP) is also totally messed up with respect to its Figure S## in the SI. In the manuscript and ToC of the SI, this is Figure S16; however, in the body of the SI, this is labeled as the second Figure S10 and discuss there as Figure S17. There is even a figure that is labeled “Figure S174”. I will not attempt to clean-up this mess, but please do so in your revision.
Answer: our apologies again; after a careful checking, various FigureS# have been modified and we feel that the corrected version is correct.
1H NMR spectrum of poly(PFP-ter-VDF-ter-HFP) is Fig. S17 whereas the 19F NMR spectrum is Figure 6 (as in the original version).
- Line 671: Please change “Of” to “of”.
Answer: this was a title of subsection and we used block letters for the first letter of all words of titles; the correction was done on the revised version.
- Line 672: Reword so you do not have passive and active verbs in the same sentence: “A study of the thermal properties of the co- and terpolymers (Table 4) showed …”.
Answer: this has been corrected.
- You have used a fairly equal mixture of “ter-polymer” and “terpolymer” as well as “ter-polymerization” and “terpolymerization” throughout the manuscript. Please correct and be consistent.
Answer: this has been corrected throughout the revised manuscript when “terpolymer” is alone. However, when it is close to « copolymers », we have mentioned “co- and terpolymer”.
- Line 721: Please italicize the “N” in “N-vinyl”.
Answer: corrected.
Our manuscript has been checked by an English speaking researcher of our Lab fluent in English writing.
Sincerely
B. Ameduri
On behalf of coauthors
Reviewer 2 Report
This manuscript investigated the copolymerization and terpolymerization of 1,1,3,3,3-pentafluoropropene (PFP) with 6 various combinations of fluorinated and hydrogenated comonomers. This type of polymerization reaction is of interest. The reactivity ranking of PFP in copolymerization and terpolymerization is compared and systematically studied. From terpolymerization, two types of products were obtained, exhibiting fair to good thermal stability under air in various solvents. However, the reviewers found that the investigation of the response was not sufficiently thorough and exhaustive. The manuscript may be accepted after a minor revision, taking into account the following four points:
1. It is preferable to discuss the reason for the introduction of PFP in the polymer at lower monomer molar masses, and the copolymer at higher molar masses (line 515 of page 15).
2. The reviewer found this difficult to understand, as CO2 dissociation involves bond cleavage, thus affording the polymer the CH2-C-CF3 radical, as shown in Scheme 5. More structural information should be provided to demonstrate the elimination of isobutylene and decarboxylation reactions. At least, a 13C NMR of the MAF-TBE should be provided to confirm the presence of isobutylene and CO2 in the polymer and an illustration of the results should be provided.
3. The solvent effect of C4F5H5 should have been mentioned in the introduction, why did most previous work use this as the solvent?
4. Are there other similar reports in the literature that fluorinated copolymers are stable? While the presence of the isocyanate group onto the aromatic ring results in a low molar mass? (line 688 of page 20)
Author Response
I thank you for your email of May 24th 2023 related to our manuscript entitled " Are (co)polymers of 1,1,3,3,3-pentafluoropropene possible?" (Manuscript ID: molecules-2398580) which included the reviewers’ comments. I appreciate their constructive remarks.
I have carefully considered all the reviewers’ remarks and suggestions and revised the manuscript accordingly. The corrected or added parts are indicated by using the yellow highlight in the revised manuscript.
Actually, a main issue was the pdf conversion that was badly done (I requested that supplied to reviewers) and though we checked the submitted pdf file, it seems that another one was supplied to reviewers (that explained the “Error! Reference source not found ..” and many other issues).
Please find below our point-by-point response to the reviewers’ comments:
This manuscript investigated the copolymerization and terpolymerization of 1,1,3,3,3-pentafluoropropene (PFP) with 6 various combinations of fluorinated and hydrogenated comonomers. This type of polymerization reaction is of interest. The reactivity ranking of PFP in copolymerization and terpolymerization is compared and systematically studied. From terpolymerization, two types of products were obtained, exhibiting fair to good thermal stability under air in various solvents. However, the reviewers found that the investigation of the response was not sufficiently thorough and exhaustive. The manuscript may be accepted after a minor revision, taking into account the following four points:
Answer: we appreciate these nice comments.
- It is preferable to discuss the reason for the introduction of PFP in the polymer at lower monomer molar masses, and the copolymer at higher molar masses (line 515 of page 15).
Answer: Actually, this study aims at finding conditions to insert as much PFP units as possible.
Because of the poor reactivity of PFP to propagate, this monomer induced some inhibition that further led to low molar mass-cooligomers.
- The reviewer found this difficult to understand, as CO2 dissociation involves bond cleavage, thus affording the polymer the CH2-C-CF3 radical, as shown in Scheme 5. More structural information should be provided to demonstrate the elimination of isobutylene and decarboxylation reactions. At least, a 13C NMR of the MAF-TBE should be provided to confirm the presence of isobutylene and CO2 in the polymer and an illustration of the results should be provided.
Answer: the release of isobutylene from MAF-TBE followed by decarboxylation have been evidenced in various articles reporting copolymers based on MAF-TBE (ref. 44, 45, 61 in the revised version) and summarized in ref. 42. Recording a 13C NMR spectrum would be a good suggestion but, as known, high concentration and accumulation are required to highlight the characteristic groups (especially for signals assigned to quaternary carbon atoms and CFx groups). In addition, the solvent used (d6-acetone) would be volatile at T> 60 °C and should affect/spoil the NMR probe if the NMR tube is not sealed. Unfortunately, no more sample is available to attempt any further analyses.
- The solvent effect of C4F5H5 should have been mentioned in the introduction, why did most previous work use this as the solvent?
Answer: We have mentioned it in bottom of page 3 that states: « both solvents (C4F5H5 or acetonitrile) being able to dissolve the fluorinated monomers without inducing any chain transfer reaction »
- Are there other similar reports in the literature that fluorinated copolymers are stable? While the presence of the isocyanate group onto the aromatic ring results in a low molar mass? (line 688 of page 20).
Answer: regarding other similar reports in the literature on the stability of fluorinated copolymers, yes, there are many in text books (see references 1, 36, 47, 50-52) or review (e.g. on VDF copolymers ref. 22), but I am not sure on the reviewer’s comment.
Regarding the presence of the isocyanate group onto the aromatic ring that results in a low molar mass, we apologize and this is not a reason. Indeed, this is mainly linked to the reactivity of the couple of monomers (or the monomer itself, knowing that the reactivity of m-TMI is much lower than that of styrene) than the presence of NCO function. For example, an alpha-fluorostyrenic (AFSt) monomer would have been much more reactive, even too reactive about PFP or CTFE and would have yielded a high AFSt content and a too low PFP (and CTFE) amount(s).
Sincerely,
Bruno AMEDURI
Directeur de Recherche au CNRS
On behalf of all coauthors